# Skeleton Action Recognition Based on Temporal Gated Unit and Adaptive Graph Convolution

Qilin Zhu , Hongmin Deng * and Kaixuan Wang

College of Electronics and Information Engineering, Sichuan University, Chengdu 610065, China
* Correspondence: hm_deng@scu.edu.cn

**Abstract:** In recent years, great progress has been made in the recognition of skeletal behaviors based on graph convolutional networks (GCNs). In most existing methods, however, the fixed adjacency matrix and fixed graph structure are used for skeleton data feature extraction in the spatial dimension, which usually leads to weak spatial modeling ability, unsatisfactory generalization performance, and an excessive number of model parameters. Most of these methods follow the ST-GCN approach in the temporal dimension, which inevitably leads to a number of non-key frames, increasing the cost of feature extraction and causing the model to be slower in terms of feature extraction and the required computational burden. In this paper, a gated temporally and spatially adaptive graph convolutional network is proposed. On the one hand, a learnable parameter matrix which can adaptively learn the key information of the skeleton data in spatial dimension is added to the graph convolution layer, improving the feature extraction and generalizability of the model and reducing the number of parameters. On the other hand, a gated unit is added to the temporal feature extraction module to alleviate interference from non-critical frames and reduce computational complexity. A channel attention mechanism based on an SE module and a frame attention mechanism are used to enhance the model's feature extraction ability. To prevent model degradation and ensure more stable training, residual links are added to each feature extraction module. The proposed approach was ultimately able to achieve 0.63% higher accuracy on the X-Sub benchmark with 4.46 M fewer parameters than GAT, one of the best SOTA methods. Inference speed of our model reaches as fast as 86.23 sequences/(second × GPU). Extensive experimental results further validate the effectiveness of our proposed approach on three large-scale datasets, namely, NTU RGB+D 60, NTU RGB+D 120, and Kinetics Skeleton.

**Keywords:** action recognition; GCN; skeleton; temporal gated unit; SE

## 1. Introduction

With the rapid development of deep learning in recent years, the topic of skeleton-based action recognition has become a focus of research. Skeleton-based action recognition has a wide range of applications in areas such as video retrieval [1], human–computer interaction [2], video image understanding [3], and more, and has attracted extensive attention from both academia and industry. Compared with the traditional RGB method, skeleton-based methods are more robust to illumination changes, camera viewpoint changes, and background noise. In addition, skeletal data, including dynamic information on simple skeletal nodes, are easier to use for training network models. The main task of human action recognition is to determine the category of a given piece of action information by feature extraction and classification. Currently, the methods used for action recognition can be classified into three types: (1) CNN-based methods [4–9], (2) RNN-based methods [10–13], and (3) GCN-based methods [14–17].

In action recognition based on skeleton data, CNN-based methods are suitable for extracting spatially correlated features and RNN-based methods are suitable for extracting temporally correlated features. GCN-based methods are becoming more mainstream for

action recognition based on skeletal data thanks to their ability to effectively characterize the spatial features of skeleton data.

With respect to CNN-based methods, Kim and Reiter [4] proposed a new temporal convolutional neural network (TCN) model for 3D human action recognition which explicitly provides interpretable spatio-temporal representations for learning and training. Li et al. [5] constructed three views in the spatial domain, making full use of temporal and spatial information to extract features; their method allows recognition scores from all views to be combined by multiple fusion methods. In [6] Yang et al. proposed a lightweight model called double-feature and double-motion network (DD-Net) based on CNN. While these CNN models have shown good performances in extracting data features in Euclidean space, they are not able to effectively extract topological information from non-Euclidean data such as skeleton data.

With respect to RNN-based methods, Song et al. [10] proposed a framework that uses an attention mechanism to learn the spatio-temporal features of skeletal data and long short-term memory (LSTM) networks to learn the relationships between adjacent frames and adjacent nodes. They employed an alternating joint training approach to train the network and designed a regularized loss function to prevent model overfitting during training. In [11], Du et al. proposed an end-to-end hierarchical RNN for skeletal action recognition by dividing human skeletal data into five parts and then feeding them into five bidirectional RNN (Bi-RNNs) sub-networks. Zhang et al. [12] designed an adaptive RNN based on the LSTM structure to automatically adjust the observation viewpoint to suit the behavior under examination. While the above RNN models have shown good performance in processing time series of information, they are not able to effectively extract topological information when dealing with skeleton data.

Finally, with respect to GCN-based methods, Yan et al. [14] proposed spatio-temporal graph convolutional networks (ST-GCN), an application of graph convolutional neural networks to feature extraction of skeleton data, and achieved remarkable results. Subsequently, many researchers have proposed improved and optimized methods based on ST-GCN. Shi et al. [15] proposed an adaptive graph convolutional network (AGCN) structure with better topology learning capability for different graph convolutional layers and end-to-end skeleton samples, which proved to be more suitable for the recognition task and its hierarchy architecture. Li et al. [3] proposed action-structural graph convolutional networks (AS-GCN) to perform action recognition by learning actional links and extending structured links. Thakkar et al. [18] proposed a part-based graph convolutional network (PB-GCN) to learn the relations among the parts of the human skeleton.

Compared with typical deep neural networks, the GCN approach provides a significant improvement in recognition accuracy. However, there are significant challenges that GCN-based methods have to face: (1) the existing GCN models are not sufficient to extract skeletal space features adaptively; (2) the existing GCN models have relatively large architectures with many parameters, which can be difficult to be trained quickly and accurately; (3) in the temporal dimension, most existing GCN models follow the temporal module in ST-GCN for temporal feature extraction. As the ST-GCN temporal module is aimed at the feature extraction of the whole sequence, non-critical frames inevitably interfere with the feature extraction of the critical frames, slowing down the learning speed and increasing the computational cost of the model.

In this paper, we first provide a detailed derivation and description of our proposed method, followed by a detailed description of the dataset and experimental configuration used. Finally, a comparison with several different mainstream models is performed in order to draw conclusions. In response to the issues mentioned above, the main contributions of this paper can be summarized as follows:

(1) The focus of this paper is devoted to improving the extraction of spatio-temporal features of skeletal sequences while considering the different importance levels of individual skeletal joints and their connections in different behaviors; here, a learnable

parameter matrix is added to enhance the capability of the model with respect to spatial feature extraction of skeletal data.

(2) A temporally gated unit is added to the feature extraction process in the temporal dimension in order to alleviate the influence of non-critical frames on feature extraction, filter out redundant temporal information, improve the inference speed of the model, and reduce computational burden.

(3) Two attention mechanisms, namely, a channel attention mechanism based on an SE module and a frame attention mechanism, are introduced in order to learn the correlations between channels and to filter out the information of critical frames, respectively.

(4) Through extensive comparison and ablation experiments, a six-layer GCN structure is proposed for spatial feature extraction, reducing the size and complexity of the model and adding residual connections to avoid model performance degradation. Extensive experiments on the NTU RGB+D 60 [19], NTU RGB+D 120 [20], and Kinetics Skeleton [21] datasets show the good performance of our proposed model.

## 2. Related Work

### 2.1. GCN-Based Skeleton Action Recognition

With the maturity of depth sensor technologies (e.g., Kinect [22]) and pose estimation algorithms [23,24], it has become possible to capture skeleton data in real time by locating key joints. Meanwhile, skeleton data have become more robust to complex backgrounds and changes in illumination, scene, color, etc. Data-driven action recognition methods based on skeleton data have attracted widespread attention and are flourishing.

Recently, this field has seen the extension of traditional convolution to graph convolution in order to better extract the topological structural information of skeleton data, and good results have been achieved. There are two main approaches for constructing GCNs, namely, the frequency domain-based approach [25–27] and the spatial domain-based approach [28,29]. The former approach uses the eigenvalues and eigenvectors of the graph Laplacian matrix for spectral analysis based on GCN, while the latter applies convolution operations directly to the vertices of the graph and its neighbors. Yan et al. [14] introduced GCN for the skeleton-based action recognition task for the first time, and proposed ST-GCN to model the skeletal data in the temporal and spatial dimensions. Based on ST-GCN, Shi et al. [15] proposed a two-stream adaptive graph convolutional network (2s-AGCN) for action recognition using the second-order skeletal information. A similar dynamic GCN proposed by Ye et al. [30] was able to provide a new global dependency modeling approach which achieved superior accuracy in skeleton-based action recognition. In [31], Chen et al. proposed a more comprehensive dual-stream GCN architecture based on graph convolutions in the vertex and spectral domains via graph Fourier transform (GFT). However, the two-stream or multi-stream structure in the above methods is slow to arrive at inferences and leads to increased computational costs and large model sizes, which represent significant obstacles when applying them to practical tasks. It remains a challenging problem to maintain or improve their recognition accuracy while reducing the complexity of the GCN model.

### 2.2. Temporal Gated Unit and SE Module

The attention-enhanced graph convolutional LSTM (AGC-LSTM) model proposed in [32] combines graph convolution with LSTM and attention mechanisms, with the gated unit in the LSTM controlling the transmission of temporal information during propagation. Inspired by this, a gated unit using LSTM is introduced in the present paper. Unlike [32], however, the gated unit introduced in this paper is utilized to filter out redundant time information, avoid disappearing gradients, and reduce computational cost. In addition, there are different correlations that exist between different channels when dealing with feature information in the time dimension. This is important in the extraction of feature

information from different channels in the temporal dimension. Therefore, we introduce an SE module to learn the correlations between channels and screen the key information.

### 2.3. Attention Mechanism

Attention modules have played an important role in neural networks and have been well studied in many different application areas, including action recognition, target detection, and natural language processing. Baradel et al. [33] proposed a novel human action recognition mechanism based on spatio-temporal attention to human postures. Song et al. [10] proposed an LSTM-based spatio-temporal attention model that automatically learns the importance levels of different nodes and different frames and weights the attention differently for each frame and node. In addition, Cheng et al. [34] embedded an attention module into their drop graph block, significantly improving its accuracy. In [35], Zhang et al. proposed a graph-aware transformer (GAT) able to make full use of velocity information to learn discriminative spatio-temporal motion features from skeleton graph sequences in a data-driven manner.

## 3. Approach

In this section, we propose a skeleton-based action recognition method using an improved GCN. We first briefly introduce the application of graph neural networks in behavior recognition. Then, we elaborate the co-operative functioning of the adaptive graph convolution module, temporal gated unit, and attention mechanism introduced in our approach.

### 3.1. Graph Convolutional Network

As mentioned above, graph convolutional neural networks are of great significance in processing graph-structured data. Skeleton data can be naturally regarded as graph-structured data. In action recognition, the bones of the human body are defined as an undirected graph in which each joint corresponds to a vertex of the graph and each bone corresponds to an edge of the graph. The skeleton sequence can be represented as a $C \times T \times N$ three-dimensional tensor, which means that there are $C$ channels, $T$ frames, and $N$ nodes. Meanwhile, a three-dimensional undirected spatio-temporal graph $G = (V, E)$ is constructed on a skeleton sequence with $N$ joints and $T$ frames, where $V = \left\{ v_i^t \mid i = 1, 2, \ldots, N; t = 1, 2, \ldots, T \right\}$ denotes the set of all joints and $E$ denotes the set of connected edges. Set $E$ consists of two parts. The first part is the connection between neighboring nodes in each frame, denoted as $E_T = \left\{ v_i^t v_j^t \mid (i, j) \in Q, t = 1, 2, \ldots, T \right\}$, where $Q$ is the set of naturally connected joint pairs in the human body. The second part is the connection between the corresponding nodes of adjacent frames, e.g., $E_F = \left\{ v_i^t v_i^{(t+1)} \mid i = 1, 2, \ldots, N; t = 1, 2, \ldots, T - 1 \right\}$.

Based on the above definition of a skeleton-based graph structure and the definition of a graph convolution operation, Yan et al. [14] constructed a multilayer ST-GCN for extracting the spatial features of the skeletal structure and redefined the formula for graph convolution, which is shown in Equation (1):

$$f_{out}\left( v_i^t \right) = \sum_{v_j \in B(v_i)} \frac{1}{Z_i^t \left( v_j^t \right)} f_{in}\left( v_j^t \right) \cdot w\left( l_i^t \left( v_j^t \right) \right) \tag{1}$$

where $f_{in}(\cdot)$ and $f_{out}(\cdot)$ denote the input and output of the feature information, respectively, $t$ denotes the $t^{th}$ frame of the skeleton sequence, $B(\cdot)$ denotes the set of neighboring nodes of node $v_i$, $w(\cdot)$ is the weight function, which provides an initial vector of weights for the input data (where the number of weight vectors is fixed while the label function $l_i^t(\cdot)$ assigns a different weight vector to each different node), and $Z_i^t(\cdot)$ is a normalization term

used to balance the importance of different neighboring nodes. In general, to implement ST-GCN, Equation (1) is transformed into Equation (2):

$$f_{out}(v_j) = \sum_{j=1}^{N} W f_{in}(v_j) \left( D^{-\frac{1}{2}} \tilde{A} D^{-\frac{1}{2}} \odot M \right) \tag{2}$$

where the annotations $f_{in}(\cdot)$ and $f_{out}(\cdot)$ have the same meaning as in Equation (1), $N$ denotes the number of vertices per frame in the skeleton data, $\tilde{A} = A + I_N$ denotes the adjacency matrix with self-connection, $I_N$ denotes an identity matrix, $\odot$ denotes the Hadamard product, and $A_{ij} = 1$ when vertices $v_i$ and $v_j$ are adjacent in physical position; otherwise, $A_{ij} = 0$, where $D = \sum_j \tilde{A}_{ij} + \varepsilon$ is a degree matrix to normalize the adjacency matrix $\tilde{A}$. To avoid zeros in $D$, we refer to [14] and set $\varepsilon$ to 0.001. Both $W$ and $M$ are learnable parameter matrices, with $W$ being the weight vector of a $1 \times 1$ convolution operation with a size of $C_{in} \times C_{out} \times 1 \times 1$ where $C_{in}$ and $C_{out}$ denote the numbers of input channels and output feature maps, respectively, and $M$ being used to adjust the importance of each edge.

Combining the above derivations, the whole process of spatio-temporal map convolution can be summarized in Equation (3):

$$f_{out}(v_i) = T_t \left( \sigma \left( \sum_{j=1}^{N} W f_{in}(v_j) \left( D^{-\frac{1}{2}} \tilde{A} D^{-\frac{1}{2}} \odot M \right) \right) \right) \tag{3}$$

where $T_t(\cdot)$ is the temporal convolution layer and $\sigma(\cdot)$ is the activation function.

In the process of spatial feature extraction, we use the method of adaptive graph convolution by adding a learnable parameter matrix to learn the differing importance of connected edges between nodes in each skeleton graph during training. To describe the adaptive graph structure, Equation (2) can be rewritten as

$$f_{out}(v_i) = \sum_{j=1}^{N} W f_{in}(v_j) \left( \Lambda_j + M_j \right) \tag{4}$$

where $\Lambda = D^{-\frac{1}{2}} \tilde{A} D^{-\frac{1}{2}}$ denotes the normalized adjacency matrix and $M$ denotes a learnable parameter matrix. The detailed structure of the adaptive graph convolution is shown in Figure 1. The size of the input feature information is $T \times N \times C_{in}$. After two paralleled and concatenated $1 \times 1$ convolutions for feature stitching, the softmax function is used to generate the learned feature matrix $M$. Finally, the learned feature matrix $M$ is summed with the original normalized adjacency matrix $\Lambda$. To avoid degradation of the model during training, we further add a residual link.

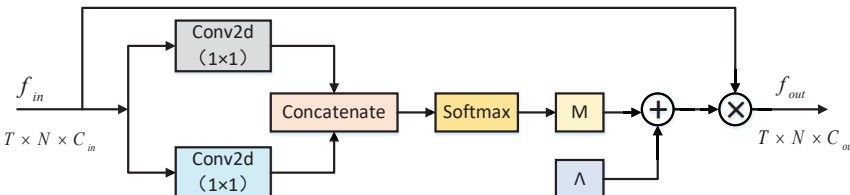

**Figure 1.** Adaptive graph convolution.

### 3.2. Temporal Gated Unit (TGU) and Temporal Convolutional Network (TCN)

Among the existing GCN-based methods, most use ST-GCN as the baseline for improvement and optimization or add additional auxiliary feature extraction modules. However, in addition to modeling the skeleton spatial information in the spatial dimension, it is important to model the temporal information in the temporal dimension. Dauphin et al. [36] first proposed a convolutional network based on gated linear units (GLU) and applied it to language modeling. While the gated unit mechanism is commonly

found in recurrent neural networks such as LSTM, the experimental results in [36] show that gated convolutional networks can help to simplify models and lead to faster convergence.

In this paper, we use a temporal gated unit for feature extraction of the temporal information of the skeleton sequence; the gated convolutional network is shown in Figure 2. These two temporal convolutional networks (TCNs) have the same kernel size $K_t \times 1$ and channel size $C$. By using TCNs and Sigmoid functions, the gated unit is able to control the transfer of time signature information between the different layers based on the contextual information of the time dimension. The output characteristic of the gated convolution depends on the TCN multiplied by the gated unit. In this way, useful information related to recognition can be retained and redundant temporal information can be filtered out. The gated convolution operation used in this method can be formulated as follows:

$$f_{gate\_out} = Sigmoid(f_{gate\_in} \cdot W + \omega) \odot (f_{gate\_in} \cdot V + \psi) \tag{5}$$

where $f_{gate\_in} \in \mathbb{R}^{K_t \times m}$ $(m = T \times N)$ and $f_{gate\_out} \in \mathbb{R}^{K_t \times n}$ $(n = T' \times N)$ denote the input and output feature information of the time gated unit, respectively, $m$ and $n$ denote the size of the input and output feature information, respectively, $W \in R^{(K_t \times m \times n)}$, $V \in R^{(K_t \times m \times n)}$, $\omega \in R^n$, and $\psi \in R^n$ denote the learnable parameters, the Sigmoid function is the activation function used to generate the gated control, and $\odot$ denotes the Hadamard product.

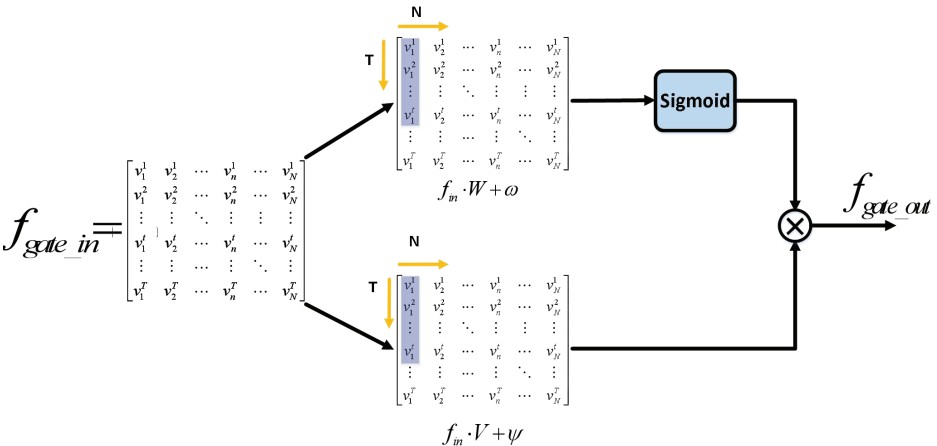

**Figure 2.** Temporal gated unit.

### 3.3. SE Block

This subsection focuses on the optimization of the SE module. The original SE module [37] is actually a channel attention mechanism indicating that features in different channels have different importance levels. In our method, we optimize the components of the SE module for use in the feature extraction module in the time dimension. The main role of the SE module is to learn the correlations between channels, providing different levels of attention to different channels while incurring only a slight increase in computational cost.

In this paper, the size of the input feature information of the SE module is $T \times N \times C$, where $T$ is the number of frames of the skeleton sequence, $N$ denotes the number of skeleton nodes in each frame, and $C$ is the number of channels of the input feature information. First, an adaptive pooling layer is used to compress the input feature information to a size of $1 \times 1 \times C$. Then, each channel is assigned a weight for output by multiplying the excitation layer with the original image. It is worth mentioning that we add residual links in this module in order to avoid gradient disappearance. In the first fully connected (FC) layer of the excitation layer, the number of channels is scaled, with $\frac{1}{r}(r \geq 1)$ being a scaling parameter intended to reduce the number of channels and thereby decrease the computational effort. In addition, we use the Tanh activation function, as the ReLU activation function used in the original SE module directly replicates feature information greater than or equal to 0 and suppresses feature information less than 0, which affects the

extraction of useful feature information by the module. The optimized SE module is shown in Figure 3. From the overall viewpoint, the number of parameters in the model is only slightly increased compared to the overall model; the increase in the number of parameters can be expressed as

$$\triangle param = 2 \times C \times C \times \frac{1}{r} \tag{6}$$

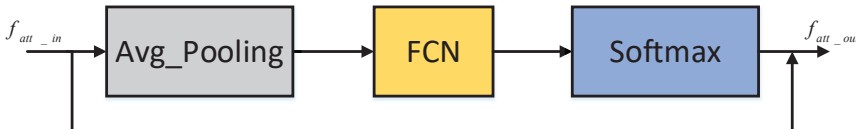

**Figure 3.** SE block.

*3.4. Attention Block*

In addition to channels, different frames may have different importance levels during the whole process of action recognition; for example, in the action of "falling", the action is similar in most frames, and changes significantly in only a few frames. It can be seen that the frames with larger changes are the key to recognizing the action. Inspired by this, we introduce an attention mechanism to enhance the weights of frames carrying key information in order to further improve the recognition accuracy of the model. In this paper, an attention mechanism based on a temporal feature extraction module is adopted. Based on the attention mechanism in [32,33], we design an adaptive weight matrix for the different frames according to the different importance levels of each frame in the whole action sequence, as shown in Figure 4. The feature information first passes through an average pooling layer to reduce the number of parameters in the attention module, then through a fully connected network to enhance the model's learning of key frames, and finally through the softmax activation function to determine the action in the key frames. To prevent the gradient from vanishing, we add a residual link in the attention module.

**Figure 4.** Attention block.

*3.5. Overall Network Structure*

In this section, we focus on the overall network framework and how each module works in terms of the overall network structure. We use a six-layer network structure, that is, six feature extraction modules cascaded to form an overall network framework; the feature extraction modules are shown in Figure 5. In the cascade of feature extraction modules, the spatial features of the skeleton sequence are always extracted first, followed by feature extraction in the temporal dimension.

The overall network framework in cascade is shown in Figure 6. The number of channels is 96, 128, and 256 in the input layer, layers 1–3, and layers 4–6, respectively. In the end, we add a classification module which contains a global pooling layer, a fully connected layer, and the softmax activation function that outputs the predicted categories. In addition, a dropout layer with a probability of 0.25 is added between the global pooling layer and the fully connected layer.

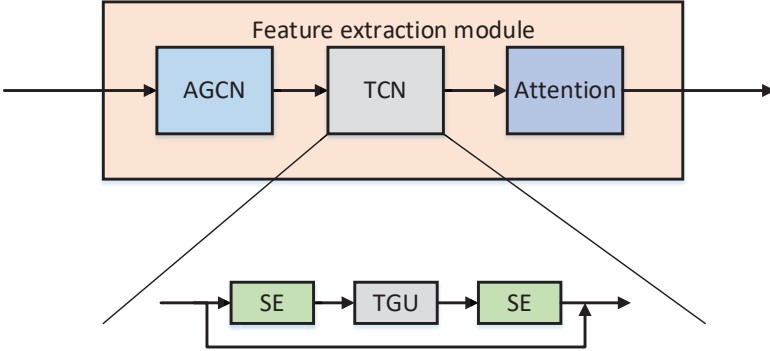

**Figure 5.** Feature extraction module.

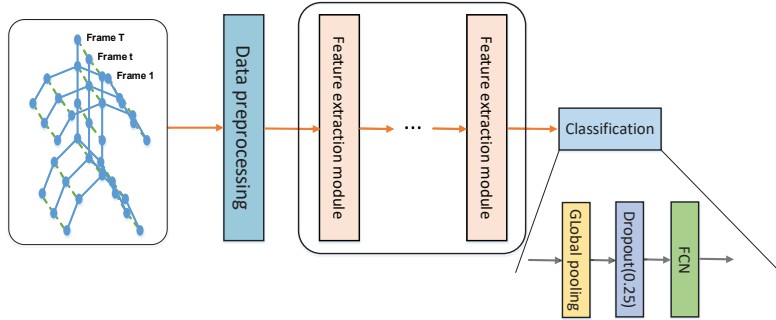

**Figure 6.** Overall network framework.

## 4. Experiments

In this section, we evaluate our model on two large skeleton-based action recognition datasets to compare it with state-of-the-art methods. We performed multiple sets of ablation experiments to evaluate the effectiveness of the additional modules used in our approach.

### 4.1. Datasets

NTU RGB+D 60 [19] is the most commonly used 3D skeleton sequence dataset for human behavior recognition, containing a total of 56,680 action clips from 60 action categories. These short films were shot by 40 volunteers in a constrained lab environment using three cameras. The provided annotations provide the 3D coordinate positions $(X, Y, Z)$ of the joint points detected by a Kinect depth sensor, with each subject's skeleton sequence containing 25 joints. In each clip, there can be up to two subjects. The two benchmarks recommended by the provider of the dataset are as follows. The cross-subject (X-Sub) contains a training set of 40,320 clips and a test set of 16,560 clips. Under this division, part of the volunteers appear only in the training set and part of them appear only in the test set. The cross-view (X-View) was captured by three cameras, with 37,920 videos captured by camera 2 and camera 3 as training set and 18,969 videos captured by camera 1 as the test set. In [20], the authors point out that 302 bad samples exist in this database, and these bad samples were removed from our experiments during training and testing. The experimental results in this paper focus on the comparison of the Top-1 testing accuracy.

NTU RGB+D 120 [20] is an expansion of the previous NTU RGB+D 60 dataset. It contains a total of 114,480 videos completed by 106 people in 155 perspectives. Similarly, there are two benchmarks. The cross-subject (X-sub120) was constructed by dividing the subjects into two groups (63,026 videos and 50,922 videos, respectively) for the training and test sets. The cross-setup (X-set120) divided the 32 IDs of the subjects into two groups of 16 each for shooting the training and testing sets, respectively. Similarly, according to [20] there were 532 bad samples, which we eliminated during training and testing.

The Kinetics [21] dataset contains 300,000 videos and 400 human action classes, with at least 400 video clips for each action. Each clip lasts for about 10 s and is taken from a

different YouTube video; this dataset only provides raw videos without skeleton annotation. Yan et al. [14] used the open source toolbox openpose to label people in each video frame as a skeleton graph structure with 18 nodes, creating the Kinetics large-scale skeleton action recognition dataset. Following the evaluation method of Yan et al. [14], the training data were set as 240,000 skeleton clips and the test data consisted of 20,000 videos. We trained our model on the training set and report the top-1 and top-5 accuracies on the test set.

### 4.2. Experiment Settings

All our experiments were conducted using the Pytorch framework and an NVIDIA GTX 1080Ti GPU. The optimization strategy used stochastic gradient descent (SGD) with the Nesterov momentum set to 0.9 and the parameters adjusted with a weight decay of 0.0001. The size of the convolution kernel was $9 \times 2$ and $1 \times 1$ in the spatial and temporal dimensions, respectively. The number of frames of each skeleton sequence was set to 300, with all 0 s filled at the end frame for each video sample with less than 300 frames. The batch size for training and testing was set to 16. A cosine learning strategy was used to adjust the learning rate during training, and the end epoch was set to 70. The channel scaling parameter $r$ was set to 4.

### 4.3. Results

We compare our proposed method with other mainstream methods in Tables 1 and 2, which show the experimental results on the NTU-RGB+D 60 dataset and on the NTU-RGB+D 120 dataset, respectively. We conducted each group of experiments three times and the standard error was no more than 0.15.

**Table 1.** Accuracy comparisons with mainstream methods on NTU-RGB+D 60 dataset. Inference speed (sequences/(second $\times$ GPU)).

| Method | Inference Speed | Param. | X-Sub | X-View |
|---|---|---|---|---|
| ST-LSTM [38] | - | - | 69.20% | 77.70% |
| Clips+CNN+MTN [7] | - | - | 79.60% | 84.80% |
| 3scale ResNet152 [9] | - | - | 85.00% | 92.30% |
| ST-GCN [14] | 42.91 | 3.10 M | 81.50% | 88.30% |
| RA-GCN [39] | 18.72 | 6.21 M | 85.90% | 93.50% |
| 2s-AGCN [15] | 22.31 | 9.94 M | 88.50% | 95.10% |
| PL-GCN [40] | - | 20.70 M | 89.20% | 90.50% |
| ST-TR [41] | - | - | 89.90% | 96.10% |
| SAGN [42] | - | 1.83 M | 89.20% | 94.20% |
| DD-GCN [31] | - | - | 88.90% | 95.80% |
| GAT [35] | - | 5.86 M | 89.00% | 95.20% |
| Proposed Method | 86.23 | 1.40 M | 89.63% | 94.91% |

In Table 1, it can be seen that the best performance of our method is 89.63% for the x-sub benchmark and 94.91% for the x-view benchmark; in addition, the number of parameters of our model is the most optimal among the compared models. Here, we compare several typical methods with each other. First, with ST-GCN [14], the most dominant backbone model based on skeletal action recognition, our accuracy is 8.13% higher than that of ST-GCN on the X-Sub benchmark, the model parameters are less than half that required by ST-GCN, and the model inference is nearly twice as fast. Second, with 2s-AGCN [15], another typical model for skeletal behavior recognition, our accuracy is 1.13% higher on the X-Sub benchmark and the number of parameters in our model is one-seventh that of 2s-AGCN. Third, compared with another novel semantic-based DD-GCN [31] method on the X-sub benchmark, our model's accuracy is better by 0.73%. Finally, compared with the latest GAT [35] model, our model has improved accuracy by 0.63% and reduces the number of model parameters by 4.46 M compared to GAT on the X-Sub benchmark. Compared with

these typical models, our model displays a more optimal structure without any reduction in accuracy.

In Table 2, it can be seen that our model achieves 84.59% and 85.64% accuracy, respectively, on the X-Sub120 and X-Set120 benchmarks. However, this accuracy is slightly lower compared to the latest model, DD-GCN [32]. The main reason for this is that a two-stream GCN structure containing vertex-domain graph convolution and graph-based spectral graph convolution is used in DD-GCN model, which has the trade-offs of an increased number of parameters and reduced inference speed.

**Table 2.** Comparison of accuracy with mainstream methods on the NTU-RGB+D 120 dataset. Inference speed (sequences/(second × GPU)).

| Method | Inference Speed | Param. | X-Sub120 | X-Set120 |
|:---:|:---:|:---:|:---:|:---:|
| ST-LSTM [38] | - | - | 55.00% | 57.90% |
| ST-GCN [14] | 42.91 | 3.10 M | 70.70% | 73.20% |
| RA-GCN [39] | 18.72 | 6.21 M | 82.50% | 84.20% |
| 2s-AGCN [15] | 22.31 | 9.94 M | 82.50% | 84.20% |
| ST-TR [41] | - | - | 81.90% | 84.10% |
| SAGN [42] | - | 1.83 M | 82.10% | 83.80% |
| DD-GCN [31] | - | - | 84.90% | 86.00% |
| GAT [35] | - | 5.86 M | 84.00% | 86.10% |
| Proposed Method | 86.23 | 1.40 M | 84.59% | 85.64% |

Table 3 compares the experimental accuracy of our model with several mainstream GCN models on the Kinetics skeleton dataset. Our model is slightly inferior to the other models in recognizing behavior in large-scale open scenes. Compared with GAT [35], our model has a 1.94% difference in Top-1 accuracy and a 1.41% difference in Top-5 accuracy. However, it is intuitively apparent from Tables 1 and 2 that our model exhibits good performance in terms of its inference speed and model parameters.

**Table 3.** Comparisons of validation accuracy with state-of-the-art methods on the Kinetics skeleton dataset.

| Method | Top-1 | Top-5 |
|:---:|:---:|:---:|
| ST-GCN [14] | 30.70% | 52.80% |
| 2s-AGCN [15] | 36.10% | 58.70% |
| ST-TR [40] | 36.11% | 58.70% |
| DD-GCN [30] | 36.10% | 59.50% |
| GAT [35] | 35.90% | 58.90% |
| Proposed Method | 33.96% | 57.49% |

### 4.4. Ablation Study

In this subsection, we verify the validity of each of the modules introduced into our model. Here, we perform experiments using the X-Sub benchmark as an example; the results are shown in Table 4, where A, B, and C denote the SE block, TGU block, and Attention block, respectively. The experimental results show that each module affects the recognition accuracy of the model to a certain extent. The recognition accuracy of the model reaches its optimum level when the three modules are combined. At the same time, we chose the category "drop" in our test to calculate the three performance parameters of precision, recall, and F-1 measure separately for comparison.

**Table 4.** The ablation experiment results for "drop" behavior.

| Method | Param. | Accuracy | Precision$_{drop}$ | Recall$_{drop}$ | F-1$_{drop}$ |
|---|---|---|---|---|---|
| Without A and B | 1.24 M | 89.33% | 0.9090 | 0.9375 | 0.9230 |
| Without A and C | 1.01 M | 88.69% | 0.9129 | 0.9258 | 0.9193 |
| Without B and C | 0.88 M | 88.58% | 0.8923 | 0.9145 | 0.9143 |
| A, B and C | 1.40 M | 89.63% | 0.9579 | 0.9479 | 0.9529 |

To verify the effectiveness of introducing the adaptive parameter matrix and the attention module, we depict a number of randomly selected samples on their attentional heat maps in Figure 7; darker colors indicate a greater contribution to the recognition of the action in the frame. Using the X-Sub benchmark as an example, we show the plot of the accuracy curve on each category of the actions in Figure 8.

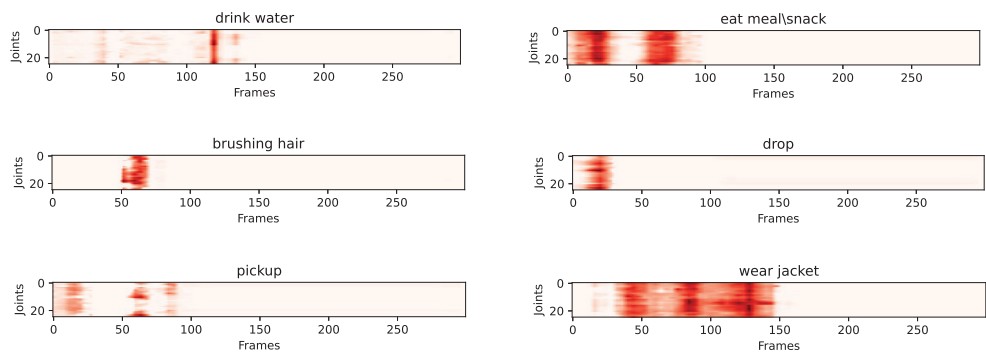

**Figure 7.** Keyframes and node heatmaps for several types of actions.

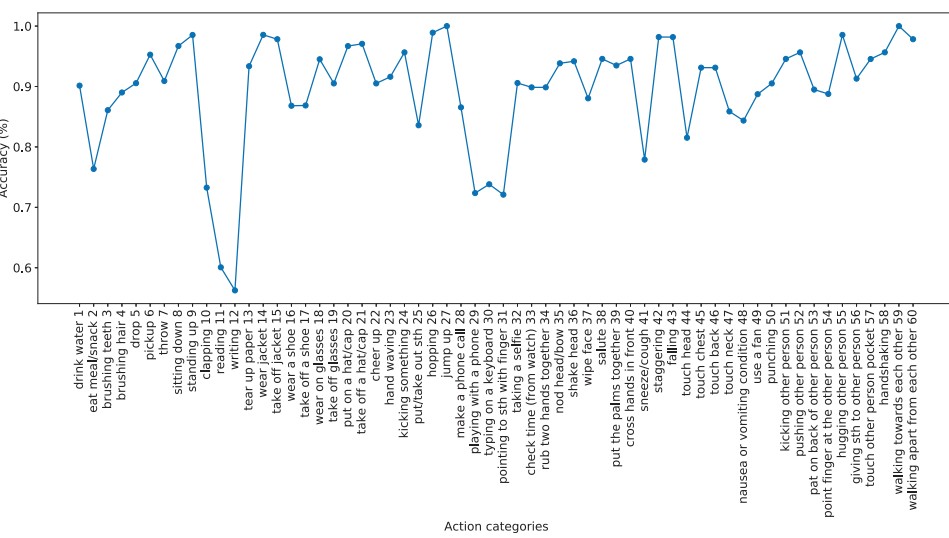

**Figure 8.** Results of accuracy comparison between each category on the NTU-RGB+D 60 X-Sub dataset; the horizontal and vertical axes denote the category and the accuracy, respectively.

## 5. Conclusions

In this paper, we propose an adaptive graph convolutional network while adding an optimized temporal gated unit and SE module to enhance the model's spatio-temporal feature extraction ability in the skeleton sequence. After extensive experiments, we finally achieve an optimal six-layer network structure model which balances the model's complexity and recognition accuracy. Of course, compared with the latest action recognition models based on skeleton data, our model has a small gap in terms of recognition accuracy. However, in terms of combined model complexity and recognition accuracy, our approach achieves the best results. In our future work, we intend to continue investigating ways

of constructing more effective skeleton graph structures to characterize small changes in actions (e.g., reading, writing, etc.), and to extend our investigation to more complex environments such as open scenes.

**Author Contributions:** Q.Z.: Conceptualization, methodology, and writing original draft; H.D. and K.W.: writing assistance. All authors have read and agreed to the published version of the manuscript.

**Funding:** This research was funded by the Natural Science Foundation of Sichuan Province, grant number 2022NSFSC0553, and partially funded by the National Natural Science Foundation of China, grant number 62020106010.

**Data Availability Statement:** All data, models, or code supporting the results of this study are available from the authors upon reasonable request.

**Conflicts of Interest:** The authors declare no conflict of interest.

## Abbreviations

The following abbreviations are used in this manuscript:

| | |
|---|---|
| GCN | Graph convolutional network |
| ST-GCN | Spatio-temporal graph convolutional network |
| TCN | Temporal convolutional neural network |
| DD-Net | Double-feature and double-motion network |
| LSTM | Long short-term memory |
| Bi-RNNs | Bidirectional RNN |
| AGCN | Adaptive graph convolutional network |
| AS-GCN | Action-structural graph convolutional network |
| PB-GCN | Part-based graph convolutional network |
| SE | Squeeze-and-excitation |
| 2s-AGCN | Two-stream adaptive graph convolutional network |
| GFT | Graph Fourier transform |
| AGC-LSTM | Attention-enhanced graph convolutional LSTM |
| GAT | Graph-aware transformer |
| TGU | Temporal gated unit |
| TCN | Temporal convolutional network |
| GLU | Gated linear units |
| FC | Fully connected |
| FCN | Fully connected network |
| SGD | Stochastic gradient descent |

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
