# Peer review of "Skeleton Action Recognition Based on Temporal Gated Unit and Adaptive Graph Convolution"

_electronics, doi:10.3390/electronics11182973_

Round 1
Reviewer 1 Report
Review Comments
In the presented work a gated temporal and spatial adaptive graph convolutional network is proposed. On the one hand, a learnable parameter matrix is added to the graph convolution layer, which can adaptively learn the key information of the skeleton data in the spatial dimension, improving the feature extraction and generalization ability of the model while reducing the number of parameters. On the other hand, a gated unit is added to the temporal feature extraction module to reduce the interference of non-critical frames to the model and reduce the computational complexity of the model. However, the following major corrections can be considered by the authors to further improve the quality of the manuscript.
I have some major corrections and suggestions below:-
1. The abstract can be improved with short text and the outcome of the work in terms of achieved performance calculations must be included in the abstract.
2. Some more recent papers based on state of art methods must be added and included as a literature review.
3. Results with respect to some more data sets must be tested and verified.
4. Comparative analysis with respect to various performance parameters must be discussed and analyzed. Various performance parameters like recall, precision, and F- 1 measure must be analyzed and discussed.
5. Future work and limitations of the proposed work can be added and discussed.
6. Performance of the proposed work on real Image datasets must be discussed. Comparative analysis with respect to inference/fps and real-time time analysis is missing.
7. How much data should be considered for training and testing for architecture implementation? Details of training and testing data sets must be tabulated.
8. Layer details of architectures must be elaborated and timing analysis needs to be added.
9. Action recognition results based on images must be discussed and included.
10. Authors must include and verify some more data sets for testing. The comparison can be a bit unfair if a different dataset is not used for comparative analysis.
11. How annotations of data have been created.
12. How is your model different and better than the proposed model and convince the work based on the novel contribution?

Reviewer 2 Report
The presented paper suggests an approach for Skeleton Action Recognition based on Temporal Gated Unit and Adaptive Graph Convolution. Overall, the work is interesting, and its organization is perfect. However, I have some additional observations to improve the quality of the manuscript further:
- Several abbreviations and acronyms are not defined.
- Abbreviations and acronyms should be defined in their apparitions.
- Authors should add a table that summarizes the abbreviations and acronyms.
- The paper contains several grammatical errors and typos. So, proofreading by a native English expert is necessary.
- The quality of Figures 7 and 8 should be improved.
- Paper’s organization should be added in the last part of the introduction.
- Authors should add some perspectives and discuss potential applications, like Face Recognition (10.1016/j.dsp.2020.102809 / 10.3390/electronics9081188).
- Authors should quantify the conclusion and the abstract (i.e., Highlight the main results of the work).
Round 2
Reviewer 1 Report
The revised version has been improved by authors who replied to my questions and took into account all my comments during the revision process.
The current version is acceptable to be published.
Reviewer 2 Report
The fourth point has not been taken into consideration.